



# How does turbulence change approaching a rotor?

Jakob Mann, Alfredo Peña, Niels Troldborg, and Søren J. Andersen

DTU Wind Energy, Technical University of Denmark

*Correspondence to:* J. Mann (jmsq@dtu.dk)

**Abstract.** For load calculations on wind turbines it is usually assumed that the turbulence approaching the rotor does not change its statistics as it goes through the induction zone. We investigate this assumption using a nacelle-mounted forward-looking pulsed lidar that measures low frequency wind fluctuations simultaneous at distances between one half and three rotor diameters upstream. The measurements show that below rated wind speed the low-frequency wind variance is reduced by up to 10% at one half rotor diameter upstream and above rated enhanced by up to 20%. A quasi-steady model that takes into account the change of thrust coefficient with wind speed explains these variations partly. Large-eddy simulations of turbulence approaching an actuator disk model of a rotor support the finding that the slope of the thrust curve influences the low-frequency fluctuations.

## 1   Introduction

It is routinely and often implicitly assumed in load calculations on wind turbines that the statistics of the turbulence does not change as the flow is approaching the rotor plane. As it is well known that the rotor affects the mean flow in front of the rotor it cannot be ruled out that also the turbulence is affected. In this paper we investigate this assumption experimentally with lidar measurements and large eddy simulation and compare the results with a simple model. We focus on low-frequency wind speed fluctuations.

Branlard et al. (2016) use vortex particle methods to calculate the effect of the turbine rotor on the incoming turbulence. They calculate the turbulent spectra at several center-line positions upstream of a Nordtank 500 kW wind turbine assuming a fixed thrust coefficient. They conclude that the presence of the rotor does not affect the turbulence spectrum significantly. However, at higher frequencies, above 0.1 Hz, they observe a slight decrease of the power spectral density when the presence of the rotor is taken into account implying marginally lower loads. They see no changes at lower frequencies. Branlard et al. (2016) emphasize that further investigations are necessary to conclude whether the effects of the stagnation on the turbulence are systematic or not (see also Branlard, 2017, which is an expanded version of Branlard's thesis).

Simley et al. (2016) measure the turbulent inflow towards a Vestas V27 wind turbine using three synchronized continuous wave, scanning Doppler lidars. They clearly see the stagnation in front of the rotor and also a slight rotation of the inflow. The standard deviation of the along-wind velocity component $\sigma_u$ decreases slightly close to the rotor plane and they hypothesize that this is linked to the reduced mean velocity, which also affects the low frequency fluctuations. They do not support this suggestion with spectral analysis and they also point out that the amount of data is limited. An additional complication is that





the Doppler lidars average the turbulent flow field in ways that depend on the direction from the lidars to the measurement volumes and their distances.

The change in the turbulence spectrum due to the stagnation in front of the rotor is investigated theoretically using rapid distortion theory by Graham (2017). In a first step, he assumes that the turbulent scales are much smaller than the size of the rotor. He also assumes that the mean flow around the rotor is described by the model of Conway (1995), a linearized actuator disk model, and that the approaching turbulence is isotropic and described by the von Kármán spectrum. With these assumptions, they derive that $\sigma_u^2/\sigma_{u\infty}^2$ increases with the induction factor $a$ of the rotor reaching a value of $\sigma_u^2/\sigma_{u\infty}^2 \approx 1.34$ at the induction factor of maximal energy extraction $a = 1/3$. Here, $\sigma_{u\infty}^2$ is the undisturbed, upstream variance of the longitudinal wind speed component $u$ and $\sigma_u^2$ is the local variance. He derives analytically that the amplification of turbulence is not equally distributed on frequencies but rather concentrated at lower frequencies leaving the inertial subrange almost unchanged. He also derives that the integral length scale of $u$ in the $y$- or $z$-direction (i.e., perpendicular to the mean flow) , which indicates how correlated fluctuations are across the rotor, increases as the flow approaches the rotor. The increase is a little less than the stretching by the mean flow in these perpendicular directions. Graham (2017) extends the theory to the more realistic case where the integral length of the turbulence is not much smaller than the rotor by concentrating on the flow along the symmetry line of the rotor. The amplification of $\sigma_u^2$ is less for the small length scale case than cases with turbulence length scales of the order of or larger than the rotor. The amplification of $\sigma_u^2/\sigma_{u\infty}^2$ decreases from 24% to 7% for $a = 1/3$ as $2L_u/D$ increases from 1 to 10, where $L_u$ is the undisturbed integral length scale of $u$ in the flow direction and $D$ the rotor diameter. The variance slowly and asymptotically approaches its upstream value as $L_u/D \to \infty$.

Farr and Hancock (2014) perform wind tunnel model studies of the flow upstream of a rotor. They find very little change in $\sigma_u$ approaching the rotor, much less than expected from the small scale rapid distortion limit discussed above. They suggest that the stagnation of the flow almost cancels out the amplification implied by rapid distortion theory.

In Sect. 2 we briefly discuss how quasi-steady fluctuations in the wind translate into fluctuations in the induction zone where we emphasize the effect of change in the induction with wind speed. That is followed by a discussion of a numerical experiment on turbulence in the induction zone in Sect. 3. Then we analyze a field experiment measuring low frequency variations in the induction zone with a pulsed Doppler lidar (Sect. 4). Finally, results are presented and discussed in Sects. 5 and 6.

## 2 Theory

The low frequency fluctuations is the focus of this paper and are discussed first. Then we summarize the results of Graham (2017), which should be valid for all frequencies but have a particularly simple solution for high frequencies.

### 2.1 Quasi-steady fluctuations

Low-frequency or quasi-steady fluctuations are defined as variations in the wind speed $U$ that are so slow that the rotor and the upstream flow has sufficient time to adjust to all the changes such that they appear as if the wind was steadily blowing at that





wind speed. If $D = 2R = 100$ m and the induction zone extends $3D$ upstream then the low frequency limit would be around $f \approx 0.03$ Hz for a free mean wind speed $U_\infty$ of 10 m/s.

For a particular wind turbine, the mean wind speed on a line extending upstream from the center of the rotor depends on the ambient wind speed $U_\infty$ and the distance from the rotor normalized by the rotor radius $\xi = x/R$ and is given by

$$5 \quad f(\xi, a, U_\infty) \equiv \frac{U}{U_\infty} = 1 - a\left(1 + \frac{\xi}{\sqrt{1 + \xi^2}}\right). \tag{1}$$

A slow fluctuation in the ambient wind speed $U_\infty$ will produce slow variations in the wind speed in the induction zone $U(x)$. The power spectral density at low frequencies is therefore amplified as

$$\frac{S(x)}{S_\infty} = \left(\frac{\partial U}{\partial U_\infty}\right)^2 \tag{2}$$

where $S(x)$ is the power spectral density (so the amplitude squared) at low frequencies at the position $x$ and $S_\infty$ is the upstream, undistorted spectrum. The partial derivative can be expanded as follows:

$$\frac{\partial U}{\partial U_\infty} = \frac{\partial f}{\partial U_\infty} U_\infty + f = f - \left(1 + \frac{\xi}{\sqrt{1 + \xi^2}}\right) \frac{\partial a}{\partial U_\infty} U_\infty. \tag{3}$$

Typically, $a$ does not change for ambient wind speeds below rated wind speed, so the second term is negligible. The spectral amplification in Eq. (2) will then be proportional to the square of relative slow down, which is of the order of but less than unity. Above rated, $\partial a/\partial U_\infty$ will become negative and a positive amplification should be seen. A similar quasi-steady model for how low frequency fluctuations of turbulence are modified by topography is presented by Mann (2000).

## 2.2 Rapid fluctuations

Rapid distortion theory for smaller turbulent scales corresponding to more rapid fluctuations is investigated by Batchelor and Proudman (1954) and Townsend (1976). Townsend calculates the response of initially isotropic turbulence to a contraction (or expansion) of the mean flow, which to some extension is what is happening in front of a rotor. The theory is used in Graham (2017) to produce amplifications of the velocity variance and the low frequency part of the velocity spectrum shown in Fig. 1.

The theory assumes that the vorticity lines are advected by the mean flow and that the approaching turbulence is isotropic as described in the Introduction and by Conway (1995). The theory implies that the amplification is strongest at the lowest frequencies and almost absent at the highest frequencies. Their results in the limit of turbulent scales much smaller than the rotor diameter are shown in Fig. 1. In a wind tunnel, the $u$-component is diminished, also relative to the other components. In contrast, the $u$-component is enhanced in the diverging flow in front of a rotor.

The novelty of Graham (2017) is that he succeeds in calculating the velocity spectrum and variance without assuming that the length scale of the longitudinal turbulence $L_{u\infty}$ is much smaller than the rotor. Graham develops the theory of Hunt (1973) further and exploit cleverly the axisymmetry of the mean flow to make the calculations feasible. As function of $L_{u\infty}/R$, the low-frequency part of the velocity spectrum decays slowly to the ambient value after a small initial increase. This is in contrast to Eqs. (2) and (3), which predict a reduction of the velocity variance equal to $f^2$ for below rated where $\partial a/\partial U_\infty = 0$. The

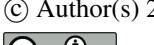



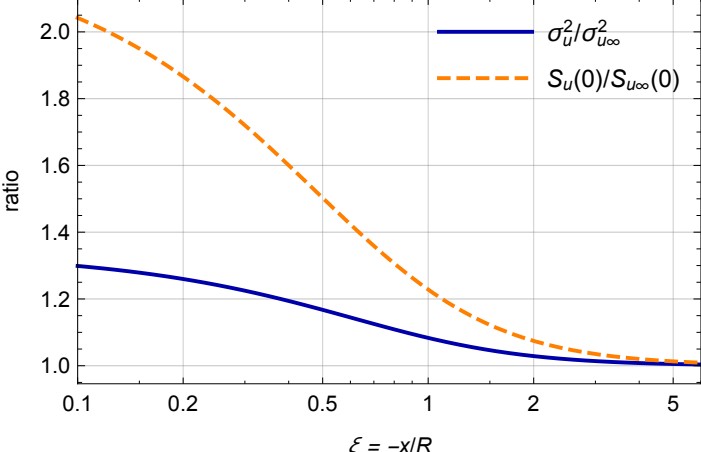

**Figure 1.** Amplification of the low-frequency spectrum and variance of longitudinal turbulence in the center of the rotor plane according to rapid distortion theory in the limit where the length scale of the turbulence is much smaller than the rotor radius. The induction factor is $a = 1/3$.

cause of this difference is that Graham (2017) does not take into account the interaction of the vorticity lines with the actuator disk. The velocity field that a vorticity line induces will be non-zero at the rotor plane, particularly for turbulence scales larger than the rotor, and the actuator disk will reduce the fluctuation caused by this vorticity line near the rotor.

## 3    Numerical techniques

5    ### 3.1    Wind turbine model

The wind turbine rotor is modelled as an actuator disk (AD) using the implementation proposed by Réthoré et al. (2014). Meyer Forsting et al. (2017) use the same model to simulate the induction zone of a 500 kW turbine and validated their predictions with lidar measurements.

The thrust force per unit area applied on the disk is assumed uniform and given by:

$$\frac{dF_T}{dA} = \frac{1}{2}\rho C_T(U_\infty)U_\infty^2 \ , \tag{4}$$

where $\rho$ is the density of air and $C_T(U_\infty)$ is the thrust coefficient as function of the free-stream velocity $U_\infty$. The free-stream velocity is the velocity that would be at the disk location if the disk was not present. This velocity is not known a priori in an unsteady turbulent setting and therefore it is convenient to express the loading of the rotor in terms of the velocity averaged over the rotor disk, $U_{\text{disk}}$. For this reason we define a modified thrust coefficient, $C_T^*(U_{\text{disk}})$, as function of the disc averaged

15    velocity

$$C_T^*(U_{\text{disk}}) = C_T(U_\infty)\frac{U_\infty^2}{U_{\text{disk}}^2} \tag{5}$$



such that Eq. (4) becomes:

$$\frac{dF_T}{dA} = \frac{1}{2}\rho C_T^*(U_{\text{disk}})U_{\text{disk}}^2. \tag{6}$$

The $C_T$ curve used in the present AD simulations is obtained from steady state simulations of the Siemens turbine presented by Troldborg and Meyer Forsting (2017) with a rated power of 2.3MW at 11.5 m/s. From their simulations, we also extract the

relation between $U_\infty$ and $U_{\text{disk}}$ and thereby the $C_T^*$ curve. Figure 2 shows the variation of $C_T$ and $C_T^*$ with respect to $U_\infty$ and $U_{\text{disk}}$, respectively. As expected $C_T^*$ reaches greater levels than $C_T$ because $U_{\text{disk}}$ is lower than $U_\infty$.

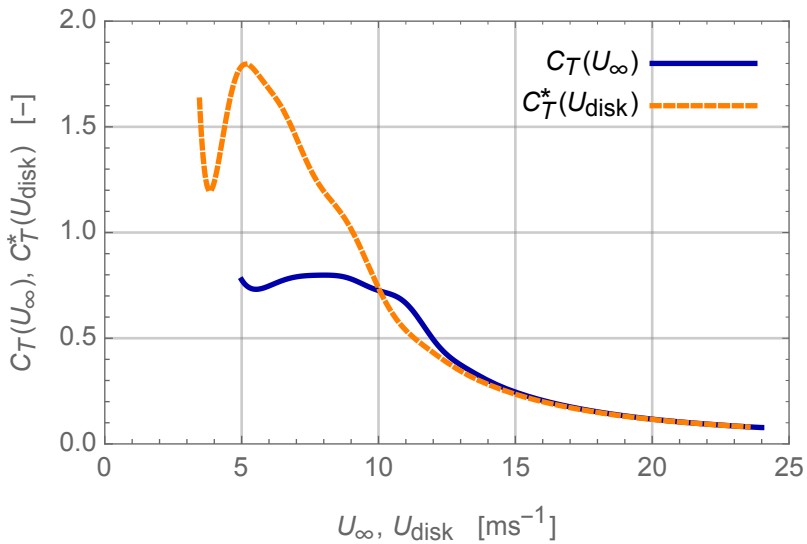

**Figure 2.** Thrust coefficient $C_T$ and modified thrust coefficient $C_T^*$ as functions of $U_\infty$ and $U_{\text{disk}}$, respectively

The loading and power of the real Siemens turbine is controlled by regulating the rotational speed and pitch of the blades. The control essentially depends on the local flow conditions at the rotor disk. Thus, using $U_{\text{disk}}$ to determine the load level at

each instant in time is a simple method for mimicking the behaviour of the controller.

### 3.2    Computational domain

The computational domain is Cartesian and has dimensions $(L_x, L_y, L_z) = (40R, 25R, 25R)$, where $L_x$, $L_y$ and $L_z$ are the domain length, width and height, respectively and $R = 46.3$ m. The rotor is located in the center of the domain, i.e. $(x, y, z) = (20R, 12.5R, 12.5R)$ with its center axis aligned with the $x$-direction (flow direction). The number of grid points in each

direction of the domain is $(N_x, N_y, N_z) = (320, 128, 128)$. In the region defined by $8.5R \leq x \leq 21.5R$, $11.05R \leq y \leq 13.95R$ and $11.05R \leq z \leq 13.95R$, the grid cells are cubic with a side length of $R/27.5$. The reason for concentrating cells in this part of the domain is to better resolve the turbulent fluctuations in the region upstream of the rotor. Outside of this region, the cells



are stretched towards the outer boundaries.

The boundary conditions are as follows: a fixed uniform velocity is prescribed at the inlet ($x = 0$), bottom ($z = 0$) and top ($z = 25R$) boundaries. Periodic conditions are applied at the sides ($y = 0$ and $y = 25R$) and a zero gradient Neumann condition is applied to the velocity at the outlet ($z = 40R$).

### 3.3 Turbulent inflow

The turbulent inflow is generated using the model of Mann (1994). The three parameters governing the Mann spectral tensor model are selected according to the findings of Peña et al. (2017), and represent the best fit to the measured conditions at Nørrekær Enge, which is the site of the lidar turbulence measurements. The output of the Mann simulation algorithm (Mann, 1998) is a spatial box of turbulent fluctuations, which are converted to time domain via Taylor's frozen turbulence hypothesis. The dimensions of the generated box are $(L_X, L_Y, L_Z) = (512R, 16R, 16R)$, with a resolution of $\Delta = R/8$.

The turbulent fluctuations are introduced into the computational domain in a cross-section located $8.25R$ upstream of the rotor using the technique described by Troldborg et al. (2014). Note, that only one quarter of the full cross-flow extent of the box is introduced in the simulations in order to avoid any influence of periodicity in the turbulence.

### 3.4 Flow solver and simulation set-up

The simulations are carried out using the incompressible Navier-Stokes flow solver EllipSys3D (Michelsen, 1992, 1994; Sørensen, 1995). EllipSys3D solves the finite volume discretized equations in general curvilinear coordinates utilizing a collocated grid arrangement. The code uses a modified Rhie-Chow algorithm (Réthoré and Sørensen, 2012; Troldborg et al., 2015) to avoid pressure velocity decoupling. The simulations are carried out as Detached Eddy Simulations (DES) using the $k - \omega$ SST (Shear Stress Transport) model by Strelets (2001). The convective terms are discretized using a hybrid scheme, which switches between the Quadratic Upstream Interpolation for Convective Kinematics (QUICK) scheme (Leonard, 1979) in the Reynolds Averaged Navier Stokes (RANS) regions and a fourth-order central difference scheme in the large eddy simulation (LES) regions. The switching is determined through a limiter function given by Strelets (2001). The coupled momentum and pressure-correction equations are solved using the Semi-Implicit Method for Pressure Linked Equations (SIMPLE) algorithm (Patankar and Spalding, 1972). The solution is advanced in time using a second-order iterative time-stepping method using a time step of $\Delta t = 0.08$ s. Simulations are carried out at free-stream velocities of $U_\infty = 7, 8, 9, 11,$ and 13 m/s, respectively in order to cover operations both below and above rated wind speed. Simulations are conducted both with and without a turbine included in the domain such that a one-to-one map in both space and time can be made of the influence of the rotor induction zone on the turbulence. The benefit of this approach is that it is insensitive to the distortion of the inserted turbulence, which is known to occur when the fluctuations are not in balance with the flow in which they are inserted.



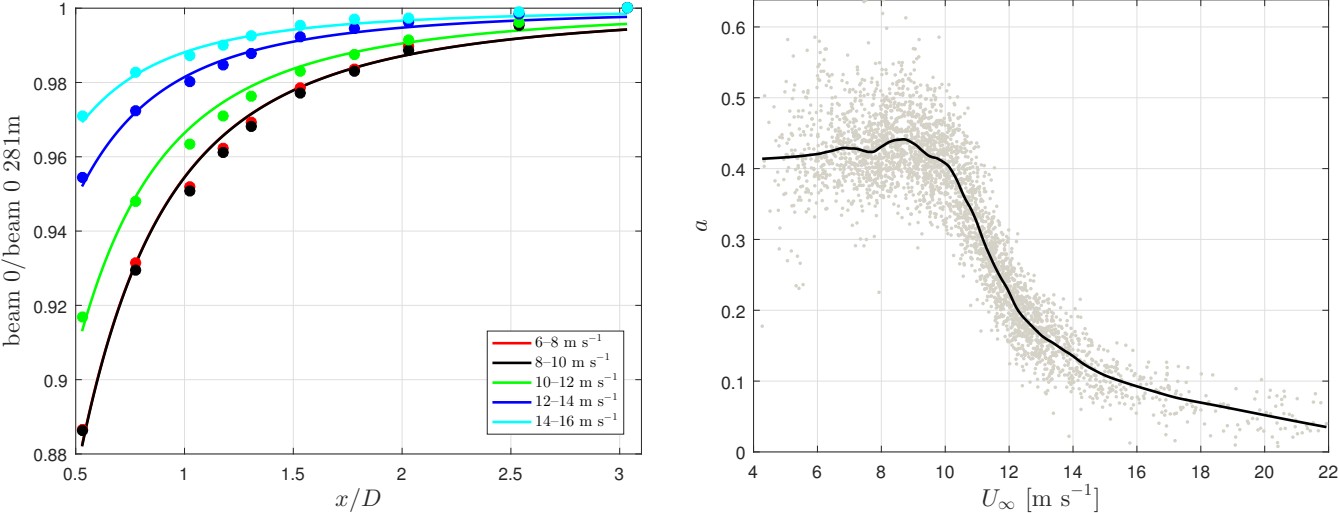

**Figure 3.** *Left:* $U(x)/U_\infty$ as a function of the upstream distance from the rotor averaged over different intervals of $U_\infty$. The curves are fits to Eq. (1). *Right:* The induction factor $a$ determined by fitting Eq. (1) to ten-minute means of the lidar measurements. The black curve is based on interval medians.

## 4 Lidar experiment

The experiment took place at a 13 wind turbine farm in northern Denmark in generally flat terrain. A five-beam pulsed proto-type lidar from Avent was mounted on the nacelle of a Siemens 2.3MW wind turbine with a hub height of 81.8 m and $D = 92.6$ m. Only the central beam of the Avent lidar looking horizontally upstream of the turbine was used in this investigation.

The lidar measured the line-of-sight velocity at ten range gates centered at 49, 72, 95, 109, 121, 142, 165, 188, 235, and 281 m upstream of the rotor at a sampling frequency of 0.2 Hz. All details about the experiment may be found in Peña et al. (2017).

## 5 Results

The line-of-sight velocity in the range gate centered around 235 m from the lidar and the wind speed from a WindSensor cup anemometer at the same distance and at hub height is compared to ensure the viability of the lidar. We find a slope deviating 1%

from one and a correlation coefficient of 0.98. The scatter is larger than other similar comparisons (see for example Sathe et al., 2015, figure 4). Due to the yawing of the turbine, the measurements are rarely collocated. An additional difference between the measurements is that the cup measures the "wind way" (Kristensen, 1999) while the lidar measures the component of the wind vector in the direction the wind turbine is pointing.

Having ensured the quality of the measurements we calculate the ten-minute average of the $u$-component of the wind and

15 fit Eq. (1) to the measurements. That gives the value of $a$ as a function of $U_\infty$, which we assume is equal to the velocity measured at the furthest range gate. The induction factors from the undisturbed sector (see Peña et al., 2017) are shown in




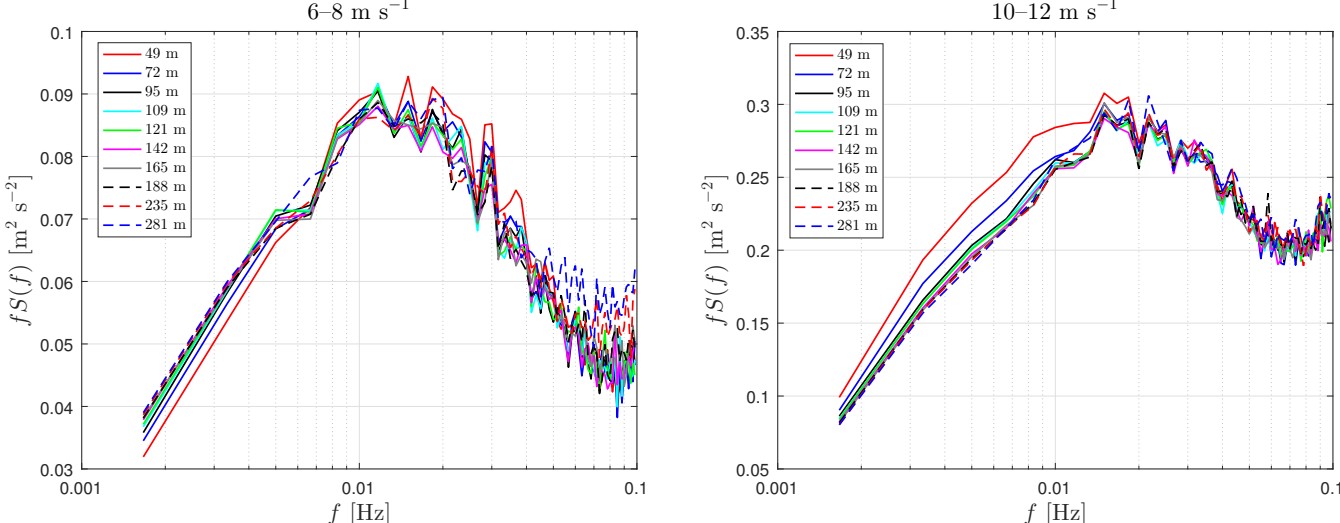

**Figure 4.** *Left:* Spectra of the line-of-sight velocity measured by the lidar as a function of up-stream distance to the rotor averaged over all measurements with $6 < U_\infty < 8$ m/s. *Right:* The same, but for $10 < U_\infty < 12$ m/s.

Fig. 3 (right) together with a smooth curve through the points, which is later used to compute $\partial a / \partial U_\infty$. For wind speeds below rated the induction factor reaches levels above 0.4 which is higher than the expected of approximately 0.3. The reason for this is that the quasi-steady model assumes a uniform load distribution and therefore tends to underestimate the induced velocity of real rotors which have a non-uniform loading, as shown by Troldborg and Meyer Forsting (2017). For a given $C_T$, this bias

causes an overestimation of the induction factor when the model is fitted to measurements of the upstream velocity. The bias is particularly dominant for the Siemens 2.3MW because it has a high local loading below rated wind speed, see Troldborg and Meyer Forsting (2017). Fig. 3 (left) shows values of the measured $u(x)/U_\infty$ averaged in intervals of $U_\infty$ with fits of Eq. (1) superimposed. It can be seen that the intervals below the rated wind speed $6 < U_\infty < 8$ and $8 < U_\infty < 10$ m/s almost coincide.

We now calculate the power spectrum of the velocity at each range gate in all 2 m/s intervals of $U_\infty$. These are based on

10-minute time series so the lowest frequency investigated is $f = 1/600\text{Hz} = 0.00167$ Hz. Two examples are shown in Fig. 4 for an $U_\infty$ where $a$ is constant with $U_\infty$ and for a velocity where $a$ rapidly decreases as a function of $U_\infty$. For low frequencies, the power spectra for the low $U_\infty$ coincide for most ranges except for those closest to the rotor when they are slightly but significantly lower. Conversely, for the higher $U_\infty$, the spectra close to the rotor are significantly higher than the upstream spectra. The experiment has the great advantage that the measurements at all range gates are done simultaneously with the

same instrument making detection of small differences possible.





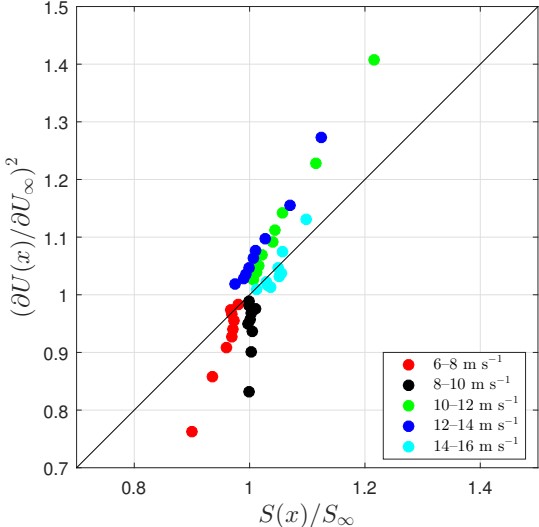

**Figure 5.** Change of low frequency ($f < 0.007$ Hz) spectral power of the longitudinal velocity component measured by the lidar compared to $(\partial U(x)/\partial U_\infty)^2$. The different points for a given wind speed interval correspond to different distances from the rotor $x$.

The experimental results are summarized in Fig. 5. Here we calculate the low-frequency spectral amplification as measured by the lidar $S_{\text{low}}(x)/S_{\text{low},\infty}$ where

$$S_{\text{low}} \equiv \int\limits_{1/600 \text{ Hz}}^{4/600 \text{ Hz}} S(f)df \quad , \tag{7}$$

i.e. we add the four lowest frequency bins of the 10-minute average spectra. We calculate the low frequency fluctuations using
different upper frequency limit. The results vary but the trend remains. On the y-axis, we plot the expected amplification according to the quasi-steady model in Eq. (3) where the induction factor and its slope is derived from the solid curve in Fig. 3 (right). The cloud of points corresponding to each $U_\infty$ bin are results from nine range gates (the tenth is used for normalization assuming it is far enough away to represent the ambient flow). The trend that the low-frequency fluctuations are reduced below rated and amplified above is captured but the exact magnitude is not.

We now turn to the analysis of the LES simulations. Since the turbulence is not completely homogeneous in the stream-wise direction, we determine the effect of the rotor on the fluctuations at a position $x$ by comparing the two simulations with and without the rotor at that position. In Fig. 6 we show the relative changes of turbulence divided into low and high frequencies. At low frequencies the fluctuations below rated wind speed are reduced while they are amplified above rated. The high frequency fluctuations, or what the LES can resolve of them, change very little.

In Fig. 7 we summarize the results and compare them with the quasi-steady model. The theoretical prediction is based on the thrust curve shown in Fig. 2. We put a fifth-order spline trough the points to be able to do derivatives and then we use the



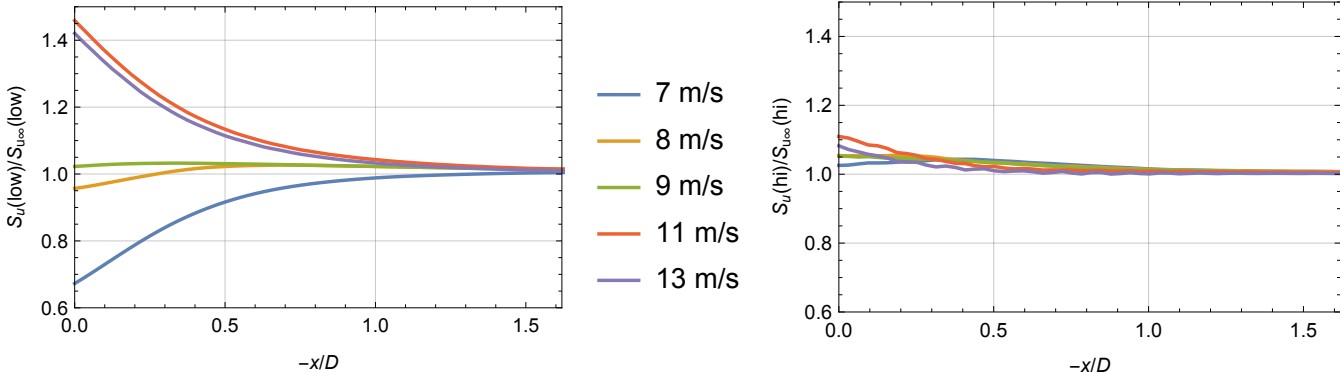

**Figure 6.** Low ($f < 0.03$ Hz) and high ($f > 0.03$ Hz) frequency variances from LES, left and right frames, respectively

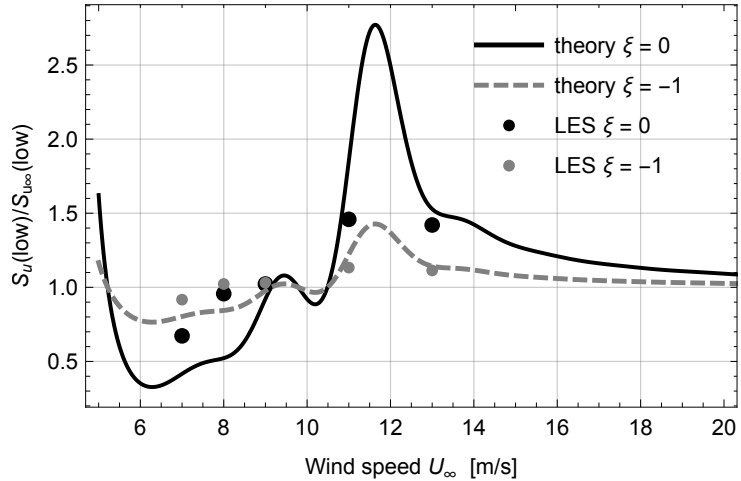

**Figure 7.** Change of low frequency ($f < 0.03$ Hz) spectral power of the longitudinal velocity component in the rotor plane and one radius upstream. The dots are the LES simulations while the lines are the quasi-steady model based on Eqs. (2) and (3)

.

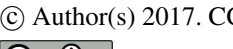

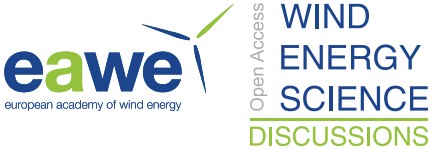

relation

$$a = \frac{1}{2}\left(1 - \sqrt{1 - C_T}\right) \tag{8}$$

to get the induction factor (Hansen, 2015). We are then able to use Eq. (3) to predict the change of low-frequency fluctuations. We do that for two distances from the rotor, $x = 0$ and $x = -R$. Again the model has the trends right, but the magnitude, especially below rated is exaggerated.

Since the theory by Graham (2017) predicts an increase of low-frequency $u$-fluctuations near the rotor, a combination of the two models could potentially improve the results.

## 6 Conclusions

The often used assumption that the statistics of turbulence approaching a wind turbine rotor is unaltered relative to its upstream values is investigated in this paper. Since the mean wind speed is reduced in the induction zone one cannot rule out that the turbulence is also affected.

A nacelle-mounted forward-looking pulsed lidar is used to measure low frequency wind fluctuations upstream of a wind turbine rotor situated in flat, homogeneous terrain. It measures wind speeds simultaneously at ten ranges between one half and three rotor diameters upstream sampling at 0.2 Hz. The integral of the velocity spectrum up to a frequency of 1/150 Hz is reduced by up to 10% at one half rotor diameter upstream and above rated enhanced by up to 20%. The changes disappear rapidly further upstream.

A quasi-steady model that uses the $C_T$-curve predicts partly the variation, but overestimates the changes. The model differs from a recent development of rapid distortion theory that is applicable also to low-frequency fluctuations (Graham, 2017).

An implementation of an actuator disk model in a large eddy simulation is used to investigate the changes in detail. The simulation is not completely homogeneous in the along-wind direction so the changes in turbulence statistics are found by comparing otherwise identical simulation runs with and without the rotor at corresponding positions. The simulations supports the finding that the slope of the thrust curve influences the low-frequency fluctuations but the simple quasi-steady model overestimates the changes. The exact consequences for loads are not investigated in this work.

*Author contributions.* JM wrote most of the manuscript except Sect. 3 which was written by NT. AP did the analysis of the lidar data, NT and SJA did the LES, JM analyzed the LES output and did the theory in Sect. 2. All authors commented on the manuscript.

*Acknowledgements.* This work is partly funded by the Unified Turbine Testing (UniTTe) project funded by The Innovation Fund Denmark (1305-00024B).





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
