# Peer review of "How does turbulence change approaching a rotor?"

_Wind Energy Science, 2017_

## Short Comment (SC1) · 7 Dec 2017

On page 1, line 22-23 you write that Simley et al. (2016) see a slight rotation of the inflow in front of the rotor.

As I understand Simley et al. (2016), they see the rotation behind the rotor. Upstream they also see a slightly positive w-component, but they explain it as "due to the gently sloping nature of the terrain between the fjord and the V27."

See also page 1-9 in "Basismateriale for beregning af propelvindmøller" (http://orbit.dtu.dk/files/53702664/ris_m_2153.pdf) which says there is no way the tangential force can affect the upsteam flow because there, in practice, is no internal friction in the air.

---

## Referee Comment (RC1) · M. Graham (Referee) · 18 Dec 2017

This is a very interesting paper as there is not much previously published showing measurements of the turbulent velocity field and spectra in the in-flow region of a full-scale HAWT together with numerical simulations which resolve the main blockage and distortion effects on the turbulent in-flow. The results are particularly interesting because they show clearly that the spectral power of the streamwise turbulent velocity component (u) at low frequencies and below-rated wind speeds where the induction factor is large, reduces significantly as the rotor disc is approached while the power at high frequencies changes much less. This is seen in both the measurements and the accompanying LES computations. In a recent paper, commented on in the present paper, [Rapid distortion of turbulence into an open turbine rotor, Graham, JFM 2017],

[Figure]

RDT theory is shown to predict a strong amplification of the spectral power of u at low frequencies as the rotor disc is approached, increasingly so the smaller the length-scale of the turbulence. At high frequencies the amplification reduces to insignificance. As observed by the authors in section 2.2 of the present paper this difference is most likely because the RDT calculation does not include the unsteady potential flow blocking effect of the rotor. This was excluded deliberately because the RDT calculations were intended to provide a correction for the incident turbulence velocity boundary condition used by lower fidelity computations which assume that the turbulence arrives 'frozen' at the rotor disc. The quasi-steady (QS) theory presented in the present paper to calculate the effects on the low frequency turbulence is an example of this and it is observed that it tends to over-predict the reduction. As is commented at the end of the present paper this may be because the amplification due to distortion is missing and that better agreement might be obtained if the RDT distortion correction were to be combined with the QS theory. The prediction of very little amplification or reduction of the spectral power of u at high frequency may be consistent similarly. Although the RDT predicts insignificant distortion in this region the unsteady potential blocking field also falls off with increasing rapidity ahead of the rotor disc for components of increasing frequency. These comments all refer to the below-rated results. It is more difficult to be sure why the power spectra of u in the in-flow clearly show considerable amplification in above-rated conditions in both the measured data and in the LES computations. In above-rated conditions the induction factor is considerably smaller and while the resulting blocking action reducing the power in the u-component is less, the effect of smaller induction factor is also to cause the distortion amplification to be much smaller. As said above this is an interesting paper presenting good quality results which I hope will stimulate further analysis on the topic of inflow turbulence.

Mike Graham, Imperial College London. 18. 12. 17

---

## Short Comment (SC2) · 9 Jan 2018

Would it be possible to add the experimental results to Figure 7?

---

## Referee Comment (RC2) · Anonymous Referee #2 · 24 Feb 2018

This is an interesting paper that pursues a clearly stated question of significant interest: how is turbulence modified at the rotor as compared to the incoming turbulence. The idea to use Rapid Distortion Theory (RDT) is solid and appropriate for the task at hand. The results presented are of interest.

An initial aspect of the paper requires clearer explanation and justification. As it is now, the initial discussion separates between slow and rapid turbulence motions, and makes the claim that RDT can be applied to the higher-frequency fluctuations but it seems to imply that RDT should not be applicable to the slow incoming, larger, slower eddies. (I am referring to the first sentence in 2.2 "Rapid distortion theory for smaller turbulent scales corresponding to more rapid fluctuations is investigated by Batchelor and Proudman.") This is contrary to the known limits of validity of RDT, which assume

that if one moves with the turbulence but the mean flow varies very, very fast compared to the turbulence, then the linearization can be justified to represent the response of the SLOW LARGE eddies. So one expects RDT to work BETTER for the slow, bigger eddies of the inflow turbulence rather than the fast ones since their intrinsic scales are slower and they cannot "nonlinearly react" to the sudden change of flow conditions. The rapid eddies (small ones) can, on the other hand, nonlinearly relax more quickly to the rapid distortion from the rotor and adjust in nonlinear fashion thus violating the fundamental conditions of RDT-validity. It is possible that the authors mean different things when they say "slow" and "large.." and "small". So, I would like to ask that they provide quantitative justification for applicability of RDT by quoting appropriately the ratio of relevant time-scales.

Another aspect that is worthwhile pointing out is to better clarify what is the relationship between this work and RDT when the mean flow undergoes rapid "Axisymmetric expansion (one contracting + 2 expanding direction). This is a tricky case for RDT and turbulence modeling, see original papers Lee 1989 Phys. Fluids A 1, 1541–1557.

Details:

In Figure 7, clearly the LES data are all around 1, and the theory too except for the peak near 11/5 m/s for xi=0. In order to more clearly compare LES to the theory, why are there no LES done for U_inf = 11.5?

---

## Author Comment (AC1) · 23 Apr 2018

**Responses to reviews of "How does turbulence change approaching a rotor?" by Mann et al. wes-2017-53**

**Jakob Mann**

**April 23, 2018**

We have received four reviews and comments which are addressed in chronological order. All changes to the manuscript are clearly marked in the attached version.

**1 Comment by M. Pedersen**

*On page 1, line 22-23 you write that Simley et al. (2016) see a slight rotation of the inflow in front of the rotor"*
*As I understand Simley et al. (2016), they see the rotation behind the rotor. Upstream they also see a slightly positive w-component, but they explain it as "due to the gently sloping nature of the terrain between the fjord and the V27." See also page 1-9 in "Basismateriale for beregning af propelvindmøller" (link) which says there is no way the tangential force can affect the upsteam flow because there, in practice, is no internal friction in the air.*

It is correct as Pedersen states that the vertical component of the flow measured in front of the rotor in Simley *et al* 2016 is not likely to be due to rotation. The flow was only measured in front of one half of the rotor and it more likely to be due to terrain effect. We have removed that statement from the paper and thank M. P. for the correction.

**2 Review by M. Graham**

*This is a very interesting paper as there is not much previously published showing measurements of the turbulent velocity field and spectra in the in-flow region of a full- scale HAWT together with numerical simulations which resolve the main blockage and distortion effects on the turbulent in-flow. The results are particularly interesting be- cause they show clearly that the spectral power of the streamwise turbulent velocity component (u) at low frequencies and below-rated wind speeds where the induction factor is large, reduces significantly as the rotor disc is approached while the power at high frequencies changes much less. This*

*is seen in both the measurements and the accompanying LES computations. In a recent paper, commented on in the present paper, [Rapid distortion of turbulence into an open turbine rotor, Graham, JFM 2017],RDT theory is shown to predict a strong amplification of the spectral power of u at low frequencies as the rotor disc is approached, increasingly so the smaller the length- scale of the turbulence. At high frequencies the amplification reduces to insignificance. As observed by the authors in section 2.2 of the present paper this difference is most likely because the RDT calculation does not include the unsteady potential flow block- ing effect of the rotor. This was excluded deliberately because the RDT calculations were intended to provide a correction for the incident turbulence velocity boundary condition used by lower fidelity computations which assume that the turbulence arrives 'frozen' at the rotor disc. The quasi-steady (QS) theory presented in the present paper to calculate the effects on the low frequency turbulence is an example of this and it is observed that it tends to over-predict the reduction. As is commented at the end of the present paper this may be because the amplification due to distortion is missing and that better agreement might be obtained if the RDT distortion correction were to be combined with the QS theory. The prediction of very little amplification or reduction of the spectral power of u at high frequency may be consistent similarly. Although the RDT predicts insignificant distortion in this region the unsteady potential blocking field also falls off with increasing rapidity ahead of the rotor disc for components of increasing frequency. These comments all refer to the below-rated results. It is more difficult to be sure why the power spectra of u in the in-flow clearly show considerable amplification in above-rated conditions in both the measured data and in the LES computations. In above-rated conditions the induction factor is considerably smaller and while the result- ing blocking action reducing the power in the u-component is less, the effect of smaller induction factor is also to cause the distortion amplification to be much smaller. As said above this is an interesting paper presenting good quality results which I hope will stimulate further analysis on the topic of inflow turbulence.*

We are happy for all the positive comments and also the consideration on combining the RDT and QS theories. We are eager to explore this combination in future work as we believe that Graham's work really has opened for future opportunities to apply RDT.

**3  Comment by A. Meyer Forsting**

*Would it be possible to add the experimental results to Figure 7?*

This is a good suggestion. We have changed figure 7 to include low-frequency spectral ratios obtained from the measurements at $\xi = -1.06$. It shows good comparison between measurements, LES and the theory.

**4  Review by anonymous Referee #2**

*This is an interesting paper that pursues a clearly stated question of significant interest: how is turbulence modified at the rotor as compared to the incoming turbulence. The idea to use Rapid Distortion Theory (RDT) is solid and appropriate for the task at hand. The results presented are of interest. An initial aspect of the paper requires clearer explanation and justification. As it is now, the initial discussion separates between slow and rapid turbulence motions, and makes the claim that RDT can be applied to the higher-frequency fluctuations but it seems to imply that RDT should not be applicable to the slow incoming, larger, slower eddies. (I am referring to the first sentence in 2.2 "Rapid distortion theory for smaller turbulent scales corresponding to more rapid fluctuations is investigated by Batchelor and Proudman.") This is contrary to the known limits of validity of RDT, which assume that if one moves with the turbulence but the mean flow varies very, very fast compared to the turbulence, then the linearization can be justified to represent the response of the SLOW LARGE eddies. So one expects RDT to work BETTER for the slow, bigger eddies of the inflow turbulence rather than the fast ones since their intrinsic scales are slower and they cannot "nonlinearly react" to the sudden change of flow conditions. The rapid eddies (small ones) can, on the other hand, nonlinearly relax more quickly to the rapid distortion from the rotor and adjust in nonlinear fashion thus violating the fundamental conditions of RDT-validity. It is possible that the authors mean different things when they say "slow" and "large.." and "small". So, I would like to ask that they provide quantitative justification for applicability of RDT by quoting appropriately the ratio of relevant time-scales. Another aspect that is worthwhile pointing out is to better clarify what is the relation- ship between this work and RDT when the mean flow undergoes rapid "Axisymmetric expansion (one contracting + 2 expanding direction). This is a tricky case for RDT and turbulence modeling, see original papers Lee 1989 Phys. Fluids A 1, 1541–1557.*
*Details:*
*In Figure 7, clearly the LES data are all around 1, and the theory too except for the peak near 11.5 m/s for $\xi = 0$. In order to more clearly compare LES to the theory, why are there no LES done for $U_\infty = 11.5$?*

Thank you for the positive comment and for pointing out that the discussion on the applicability of RDT is half misleading and half absent. We agree that careful assessment of the temporal and spatial scales should be performed in this paper.

Therefore, we have added a detailed explanation section 2.2 and also changed the header of that section (see the attached new version of the manuscript). It shows that only eddies that are smaller then a meter or so are not effectively "frozen" while experiencing the distortion by the induction zone.

We have included a reference to Lee (1989) stating that his results are in principle included in the analysis by Graham (2017) and they also do not take into account the interaction of the vorticity and the rotor because it is absent in his analysis.

Finally we have conducted yet an LES for $U_\infty = 11.5$ m/s as suggested by the reviewer and included the results in figures 6 and 7. The simulations indeed

peak around 11.5 m/s although at a much lower value.

[revised manuscript text omitted]